# Incremental Numerical Approach for Modeling the Macroscopic Viscoelastic Behavior of Fiber-Reinforced Composites Using a Representative Volume Element

**DOI:** 10.3390/ma15196724

**Published:** 2022-09-27

**Authors:** Nicolas Gort, Igor Zhilyaev, Christian Brauner

**Affiliations:** Institute of Polymer Engineering, FHNW University of Applied Sciences and Arts Northwestern Switzerland, Klosterzelgstrasse 2, 5210 Windisch, Switzerland

**Keywords:** composite process simulation, relaxation, homogenization, thermoset, residual stress

## Abstract

The objective of this study is to describe the stress relaxation behavior of an epoxy-based fiber-reinforced material. An existing incremental formulation of an orthotropic linear viscoelastic material behavior was adapted to Voigt notation and to the special case of an isotropic material. Virtual relaxation tests on a representative volume element were performed, and the behavior of individual components of the relaxation tensor of the transversely isotropic composite material was determined. The study demonstrated that in the case of only one viscoelastic material, each component of the relaxation tensor can be described in terms of a scalar form factor and the behavior of the neat resin. The developed method was implemented in an incremental finite element model (FEM) analysis to calculate the stress relaxation on the macroscopic ply level. A validation of the approach has shown a promising agreement up to a limit below the glass transition temperature of 15 °C in longitudinal and 35 °C in transverse direction. This study therefore demonstrates a novel way to incrementally describe the macroscopic viscoelastic behavior of materials with a single viscoelastic component with good controllability for engineering purposes.

## 1. Introduction

Fiber composite materials are known to have the potential to replace other materials in areas subjected to high mechanical loads. These materials can be particularly attractive for saving weight in moving masses. However, their potential can only be exploited if knowledge gaps in production and use are closed. This was the aim of the project within which this study was conducted. The research activities were focused on the influence of residual stresses on the fatigue strength of a composite leaf spring for the automotive industry.

During the manufacturing process of a thermoset material (e.g., compression resin transfer molding CRTM), its structure is exposed to chemical shrinkage and thermal contraction. Depending on the geometry of the part, especially if free deformation is restricted, this can lead to the formation of process-induced stresses. Analogous to a mechanical relaxation test, these stresses will relax with time, especially at higher temperatures and a low curing degree. In a typical manufacturing process for fiber-reinforced polymers, the material is in exactly this state at the beginning of the process, which means at comparably high curing temperatures compared to the actual glass transition temperature. When one estimates the extent of the residual stresses, the stress relaxation has to be considered in order not to make an overly high and thus conservative prediction.

For this reason, the introduction of a homogenization model to describe the relaxation effect of the composite based on the behavior of its individual components is advantageous. Isolated measurements of the components of the relaxation tensor are difficult to perform since a unidirectional strain is challenging to produce in a mechanical test. In contrast, an isotropic material can be described in terms of only two parameters, so the elimination of the lateral contraction is not necessary. Therefore, the relaxation behavior can also be measured in a multiaxial strain state, e.g., uniaxial tension. Furthermore, the introduction of a homogenization model from an isotropic to an orthotropic material allows the consideration of different fiber volume contents.

Within the current project, a virtual process simulation was set up based on earlier models [1,2], in which the modeling of the relaxation effect was the last in a series of implemented physical effects. It was assumed that for some loads, a lack of knowledge could lead to significant inaccuracy in the predicted final performance. To remedy this knowledge gap, the present study was performed, which investigated the magnitude of relaxation of isolated loads on microscopic unit cells. Its aim was to derive a method that can adapt to varying properties and contents of the individual components of a composite and is able to provide input values for the manufacturing process simulation implemented in an incremental FEM analysis.

### 1.1. State of the Art

The available approaches used to model process-dependent residual stresses in fiber-reinforced materials show a trend in which increasing accuracy comes with increased complexity and computational effort.

The most drastic simplification of the manufacturing process is an elastic model that considers the total thermal and/or chemical shrinkage together with the stiffness at a reference temperature, usually room temperature. The amount of thermal shrinkage is thus calculated as the difference between the reference temperature and a defined stress-free temperature (usually the curing temperature) multiplied by a coefficient of thermal expansion [3]. Since the material’s stiffness at the curing temperature is lower than at room temperature, this approach obviously leads to an overestimation of the stresses that occur in reality. This has the beneficial effect that an assessment of a manufactured part is conservative, which is generally required for engineering purposes.

To increase the accuracy, incrementally elastic models, for example, the CHILE (cure hardening instantaneously linear elastic [4]) model, have been developed. They are especially popular due to their ability to be easily implemented in an incremental FEM analysis [5,6]. This approach derives the occurring stress from the integration of the occurring strain with the actual stiffness over time: (1)$$\sigma \left(t\right)=\int_{0}^{t}E\left(T,\alpha \right)\frac{d\varepsilon }{d\tau }d\tau$$

A further increase in the accuracy of the predicted stresses can be achieved if the viscoelastic effects are considered. It was found in [7] that the elastic approaches overestimate the occurring stresses up to 20% in some specific cases.

In linear viscoelastic approaches, the elastic modulus is replaced with a time-dependent modulus so that the formulation becomes: (2)$$\sigma \left(t\right)=\int_{0}^{t}E\left(t-\tau \right)\frac{\partial \varepsilon }{\partial \tau }d\tau$$

This integral, known as the convolution integral, takes an easier form if it is transformed into the Laplace domain. An example of viscoelastic approaches that take advantage of the Laplace transform can be found in [8], where a material model for woven fabrics was developed. The validation was carried out for the distortion of a plate with an asymmetrical layout. Other approaches, especially for the case of epoxy resin systems, were developed in [9,10].

A drawback of the methods involving Laplace transformation is that they are computationally expensive and because, in some cases, no analytical solution for the back-transformation can be found. To avoid these problems, approaches in the time domain have also been developed. Zocher et al. [11] developed an incremental formulation for a linear viscoelastic material whose stiffness tensor entries can be described in terms of the rheological model known as the generalized Maxwell model. Since this model is known to provide a good approximation for the relaxation behavior of a wide range of materials and at the same time it is possible to be implemented in an incremental FEM analysis, this approach is very promising. An example of the application of this formulation can be found in [12], where the material was modeled to relax instantly in the rubbery phase and show almost elastic behavior in the glassy phase during the curing of an epoxy-based thermoset composite material.

The presented viscoelastic approach was integrated in a holistic approach of a virtual manufacturing process [1]. It aims to include all relevant physical aspects leading to process-induced stresses and consists of a thermochemical and a subsequent mechanical part. As mentioned earlier, one area of uncertainty is the so-called virtual material characterization of the relaxation properties of a composite material from the properties of its individual components. While there are well-established concepts for elastic homogenization, there are few methods available for viscoelastic homogenization that are user-friendly and allow a fast assessment of the accuracy. In the following, we first present the available methods for homogenization of the elastic properties and then those for the viscoelastic properties.

#### 1.1.1. Homogenization of Elastic Properties

In comparison with viscoelastic behavior, there are several widely used homogenization concepts for elastic behavior. They can be differentiated into empirical, analytical and numerical approaches.

A well-known empirical relationship was developed, namely the Halpin–Tsai equations [13,14]. All the mechanical properties can be described in terms of two equations as follows:(3)$$\frac{p}{p_{m}}=\frac{1+\xi \eta v_{f}}{\left(1-\eta v_{f}\right)}E_{m}\;where\;\eta =\frac{p_{f}/p_{m}-1}{p_{f}/p_{m}+\xi^{\prime }}$$
where *p* is an arbitrary property of the composite, *p_m_* and *p_f_* denote a property of the matrix and fiber, respectively, and the parameter *ξ*′ is an adjustment parameter that needs to be determined to fit to the experimental data. In order to correct its accuracy for shear stiffness, ref. [15] formulated a correction factor for this formulation.

Analytical approaches usually use drastic simplifications of the composite material. A well-known approach is the rule of mixtures, which calculates the homogenized stiffness from differently arranged springs representing the stiffness of the matrix and the fiber, respectively. By arranging the springs in parallel, it is possible to describe the fiber-dominated properties such as the stiffness in the fibers’ direction, and arranging the springs in series can be used to describe matrix-driven properties such as the transverse stiffness or shear stiffness. A more accurate micromechanical model based on this concept is known as the rule of mixture developed by Chamis [16], which rose to eminence in the aerospace industry.

In terms of numerical homogenization, three approaches are commonly used for simulating composites: a micromechanics-based approach, an equivalent homogeneous material (EHM) based approach and a combination of the first two methods. Each approach has advantages and disadvantages described in the literature [17,18,19,20]. The micromechanical approach describes the material structure in detail, and it is possible to investigate local defects [18]. However, the computational effort is very high, because a much better mesh quality is required in comparison to the EHM model. The EHM method reduces computational effort by homogenizing the microscopic properties on the macro level, but it neglects local effects [21,22]. It is possible to combine the two models [18,20,23] to receive the advantages of both.

Microscale modeling involves consideration of uncertainties in the composite structure, which requires stochastic models [24]. In microscale models the composite properties are defined from all constituent materials. Basic approaches include elasticity theories based on the repetition of a representative unit cell, assuming a perfect bond between fibers and matrix. However, an advanced viscoelastic models could be implemented to describe fiber-media interaction and the fiber packing process [25]. The focus of the micro-scale study is on the fiber composition, geometry, and orientation. Such micromechanical theories are usually validated with experiments [24,26,27].

Multiscale techniques solve local problems on the microscale and take this information to the macro level through homogenization techniques [28]. The material properties of the macroscale model could be obtained from micromechanical analyses. Mechanical behaviors of heterogeneous materials are often described with representative volume elements (RVE) [29]. Hill’s theory assumes that the RVE should contain enough fibers to provide a statistical representation of the heterogeneous materials. One advantage of the RVE method is that the composite material replaced by a homogeneous media retains the anisotropic material properties. The effective properties derived from the RVE represent the material on the macroscopic scale, which is commonly known as the micro–meso–macro principle [30].

#### 1.1.2. Homogenization of Viscoelastic Properties

Consideration of the time-dependent behavior of the individual components and their interaction brings additional complexity into the homogenization of a composite‘s properties. A common approach uses the principle that if the convolution integral in Equation (2) is transformed into the Laplace space, it takes the form of basic multiplication. Afterwards, a micromechanical homogenization approach (as well as classical methods from elastic homogenization) can be applied for a specified fiber volume content:(4)$${\hat{\Gamma }}_{c}\left(s\right)=MM\left({\hat{\Gamma }}_{f}\left(s\right),{\hat{\Gamma }}_{m}\left(s\right),v_{f}\right)$$

Here, *MM* denotes the micromechanical homogenization model, $\hat{\Gamma }$ the viscoelastic response in Laplace space of the fiber, matrix, and composite respectively and $v_{f}$ the fiber volume fraction.

Another common approach for the homogenization in Laplace space is known as the asymptotic homogenization method (AHM). It considers the composite as a periodic assembly of RVE’s. One of the first applications of this method was carried out by [31]. With a focus on the individual indices of the creep or relaxation tensor, an early evaluation was presented in [32], where a micromechanical model was also developed directly in the Laplace domain. A more recent example of this method was given by [33], where a Prony’s series was used in order to model the individual phases’ relaxation in the time domain. The results for low fiber volume contents can be found in a later study [34].

As mentioned earlier, the main challenge of the application of Laplace transform is to obtain analytical formulas for the inverse Laplace transform to generate the viscoelastic solution back in the time domain. Therefore, a drawback of this method is the mathematical complexity which limits its wide application by different users.

For this reason, other researchers have applied homogenization in the time domain. In [35], quasi-elastic solutions were developed for cases of elementary loads summarized in a k-factor for each of them. While these factors can be used to describe the time-dependent behavior of the creep tensor indices, they cannot be used for the relaxation tensor indices. This limitation arises because there are no widely used elementary load cases for producing a unidirectional strain, since contraction in the lateral direction is hard to restrict in a mechanical test. However, it is remarkable that some indices increase over time, which is also shown in the current investigation. A similar approach was recently followed by [36], who decomposed the strain tensor to represent characteristic load cases. A solution for determining the stress relaxation of the composite material for these load cases was developed and validated. Although it is a very promising approach, it is hard to implement due to the high number of parameters and solution uncertainty for specific load cases.

#### 1.1.3. Limits of the Linear Viscoelastic Assumption

The assumption that linear viscoelastic material behavior describes the homogenized composite’s behavior might be inaccurate because local stresses are much higher on the microlevel compared with the meso-level or ply level. On the microscale, the material shows greater stress by a factor of 3–12 depending on the stiffness ratio between the polymer and the fiber. It might exceed linear behavior (an assumption for the application of linear viscoelasticity) even at low load levels.

This is especially visible in the transverse direction. Brauner et al. [37] simulated the stress distribution in a unit cell by considering thermal and chemical shrinkage, temperature-dependent effects, micro-yielding, and degradation and relaxation. An example considering just thermal strain is shown in Figure 1. An applied nominal strain of 0.48% can lead to a maximum local strain of 2.93%, which is an amplification of about 6.

### 1.2. Approach

In this study, an incremental formulation of the linear viscoelastic response [11], was used to analyze the response of isolated components of the orthotropic relaxation tensor with the use of an RVE.

It has been suggested that in the case of only one time-dependent material, the response can be described as proportional to the response of that material because the decay time may not change. For this reason, the possibility of expressing the resulting time-dependent behavior of the individual indices in terms of a scalar form factor, together with the behavior of the neat resin system, was investigated.

These factors were integrated in [11] in order to provide an incremental description of the composite material. The presented approach was implemented by using user subroutines in the commercial finite element program Abaqus (6.13, Maastricht, The Netherlands). A validation was performed by measuring an epoxy-based glass fiber reinforced composite material, demonstrating its simplicity and computational advantage. Even if the linear viscoelastic region might be exceeded at the microscale, the suitability of the homogenization approach assuming this behavior can be assessed by the validation on the macroscale.

The present study therefore provides a novel approach for the description of a composite with only one linear viscoelastic phase, which is applicable to a wide range of materials. The form of expressing the elements of the relaxation tensor in terms of form factors offers a good controllability because each factor indicates the deviation of the response of the element to which it belongs to the response of the neat resin system. The incremental form of this description in the time domain also represents an extraordinarily efficient variant compared to established methods using Laplace transformation, since only six status variables have to be transmitted for each time step.

## 2. Materials and Methods

The research flow of the present work was structured according to the proposed approach: First, the formulation of [11] was adapted for implementation in a commercial finite element program (Section 2.1). A description for the orthotropic case has been simplified to the isotropic case, which can represent a viscoelastic resin system (Section 2.2). The elastic and viscoelastic properties of the resin system were determined in a three-point bending test so that this data could be used to calibrate the viscoelastic phase of an RVE (Section 2.4). Next, the response of the elements of the relaxation tensor to isolated load cases was determined and it was evaluated whether these can be described using a factor and a master curve (Section 2.3). Since this turned out to be promising, a formulation for describing the macroscopic behavior from these factors was developed and a validation was carried out to estimate the accuracy of the approach. For this purpose, a virtual copy of the 3PB experiment was created (Section 2.5) and compared with experimental data from the composite (Section 2.4).

### 2.1. Incremental Orthotropic Linear Viscoelastic Formulation

The general stress–strain relationship for an orthotropic linear viscoelastic material can be expressed by the constitutive equation: (5)$$\sigma_{ij}\left(x_{k},\xi \right)=\int_{0}^{\xi }C_{ijkl}\left(x_{k},\xi -\xi^{\prime }\right)\frac{\partial \varepsilon_{kl}\left(x_{k},\xi^{\prime }\right)}{\partial \xi^{\prime }}d\xi^{\prime }$$

This respects the history of the material by assuming that each response to a loading event can be superposed on all other responses. A common method for the description of the relaxation behavior of a linear viscoelastic material is to use a rheological model. One such model known to be suitable for describing the relaxation behavior of a wide range of materials is named the generalized Maxwell model, also known as Wiechert model [38]. It consists of a number of so-called branches, each consisting of a spring and a dashpot, which are connected in parallel (Figure 2).

The generalized Maxwell model has the beneficial property that the stress it describes during a relaxation test takes the form of a Prony series. Now, by assuming that the behavior of the elements of the stiffness tensor in Equation (5) can be represented in terms of a generalized Maxwell model with M branches, its value at time $\xi_{n+1}$ (following the notation of [11].) can be expressed as a Prony series as follows: (6)$$C_{ijkl}\left(x_{k},\xi_{n+1}-\xi^{\prime }\right)=C_{ijkl_{\infty }}+\overset{M_{ijkl}}{\underset{m=1}{\sum }}C_{ijkl_{m}}e^{-\left(\xi_{n+1}-\xi^{\prime }\right)/\rho_{ijkl_{m}}}$$

Comparing this description with the rheological model, one can see that spring elements that are connected to a dashpot contribute the less to the actual stress the longer it has been that strain was applied. This is commonly known as fading memory behavior.

Equation (6) was inserted in Equation (5) and converted to an incremental form by [11]. That means that the actual stress is the sum of the stress increment and the stress of the previous time step:(7)$$\sigma_{ij\left(t+\Delta t\right)}=\sigma_{ij\left(t\right)}+\Delta \sigma_{ij}$$

It consists of an immediate increment, a stress increment due to thermal strain and a stress increment ($\Delta \sigma_{ij}^{R}$) that considers relaxation of the stress. For small time steps, the immediate increment can be understood as the elastic increment and $\Delta \sigma_{ij}^{R}$ as the viscoelastic part.
(8)$$\Delta \sigma_{ij}=C_{ijkl}^{\prime }\Delta \varepsilon_{kl}-\beta_{ij}^{\prime }\Delta \Theta +\Delta \sigma_{ij}^{R}$$

In contrast to [11], here the temperature load considered (ΔΘ) was integrated with the mechanical load, $\Delta \varepsilon_{kl}$. The simplified incremental description of linear viscoelastic behavior then takes the form (9)$$C_{ijkl}^{\prime }\equiv C_{ijkl_{\infty }}+\frac{1}{\Delta \xi }\overset{M_{ijkl}}{\underset{m=1}{\sum }}\eta_{ijkl_{m}}\left(1-e^{-\Delta \xi /\rho_{ijkl_{m}}}\right)\;\left(\mathrm{no}\;\mathrm{sum}\;\mathrm{on}\;i,j,k,l\right)\;$$
(10)$$\Delta \sigma_{ij}^{R}=-\overset{3}{\underset{k=1}{\sum }}\overset{3}{\underset{l=1}{\sum }}\overset{M_{ijkl}}{\underset{m=1}{\sum }}\left(1-e^{-\Delta \xi /\rho_{ijkl_{m}}}\right)S_{ijkl_{m}}\left(\xi_{n}\right)\;\left(\mathrm{no}\;\mathrm{sum}\;\mathrm{on}\;i,j,k,l\right)\;$$
(11)$$\begin{array}{cc} S_{ijkl_{m}}\left(x_{k},\xi_{n}\right)= & e^{-\Delta \xi /\rho_{ijkl}m}S_{ijkl_{m}}\left(x_{k},\xi_{n}-\Delta \xi \right)\\ & +\eta_{ijkl_{m}}R_{\varepsilon ,kl}\left(1-e^{-\Delta \xi /\rho_{ijk_{m}}}\right)\;\left(\mathrm{no}\;\mathrm{sum}\;\mathrm{on}\;i,j,k,l\right)\; \end{array}$$
where:(12)$$R_{\varepsilon ,kl}\equiv \Delta \varepsilon_{kl}/\Delta \xi$$

The variable *S* represents a state variable that must be stored in the previous time step and passed to the current step. For the orthotropic case, each branch m of the Maxwell model requires a state variable for each index of the stress tensor *ij*. This leads to M × 36 state variables, which is cumbersome for practical use.

For presenting the result of fitting the rheological model to the measurement data, it is beneficial to introduce a parameter that describes the relationship of one spring element to the sum of all spring elements:(13)$$\varphi_{m}=\frac{k_{m}}{k_{\mathrm{tot}}}$$

### 2.2. Adaption to Isotropic Material Behavior

The homogenization for modeling a unit cell was carried out by considering the individual phases as isotropic and only the matrix’s material properties to be time-dependent. This was achieved by expressing the stress–strain relation in terms of the bulk and shear modulus, where only the shear modulus was time-dependent, similar to the approach of [39]. This means, that we can assume that the viscous part of the deformation is incompressible and thus the volumetric change is purely elastic. Hence, the constitutive stress–strain relationship is written as:(14)$$\sigma_{ij}=K\varepsilon_{el,vol}I+2G\varepsilon_{d,ij}$$

Here, the strain in the first term of the right-hand side is the volumetric change, derived from the trace of the strain tensor as:(15)$$\varepsilon_{el,vol}=trace\left(\varepsilon_{el}\right)$$

The second is the deviatoric strain, which is defined as:(16)$$\varepsilon_{d}=\varepsilon_{el}-\frac{1}{3}\varepsilon_{el,vol}I$$

If we consider the shear modulus to be time-dependent; it can be written in terms of the linear viscoelastic constitutive equation. The stress–strain relationship for an isotropic material can be described by: (17)$$\sigma_{ij}=K\varepsilon_{el,vol}I+2\int_{0}^{t}G\left(t-t^{\prime }\right)\frac{\partial \varepsilon_{d,ij}}{\partial t^{\prime }}dt^{\prime }$$

Therefore, the shear modulus is derived from the measurement results by assuming the bulk modulus to be constant:(18)$$G\left(t\right)=\frac{3KE\left(t\right)}{9K-E\left(t\right)}$$

Similar to the elements of the stiffness tensor in Equation (6), it is possible to describe the relaxation behavior of most polymer materials with the generalized Maxwell model and thus in terms of a Prony series as follows: (19)$$G\left(x_{k},\xi_{n+1}-\xi^{\prime }\right)=G_{\infty }+\overset{M}{\underset{m=1}{\sum }}G_{m}e^{-\left(\xi_{n+1}-\xi^{\prime }\right)/\rho_{m}}$$

Inserting the constitutive equation of the isotropic material in Equation (17) and the Prony series in Equation (19) in the procedure of Zocher [11], the general incremental form (Equations (8)–(11)) can then written in simplified form as:(20)$$\Delta \sigma_{\alpha }=K\Delta \varepsilon_{el,vol}I_{\alpha }^{V}+2G^{\prime }R_{\alpha }^{-1}\Delta \varepsilon_{d,\alpha }+\Delta \sigma_{\alpha }^{R}$$
where $I_{\alpha }^{V}$ denotes the identity tensor in Voigt notation (Appendix A). Similarly, *R* is a transformation vector from index to Voigt notation. In Equation (2D), it is also known as the Reuter matrix (Appendix A). The variable *G*′ is defined as: (21)$$G^{\prime }\equiv G_{\infty }+\frac{1}{\Delta \xi }\overset{M}{\underset{m=1}{\sum }}\eta_{m}\left(1-e^{-\Delta \xi /\rho_{m}}\right)$$

This term can be linearized while maintaining the conservative character of this approach. For small time steps applying the rule of de L’Hospital, it takes the form of the shear modulus: (22)$$\underset{\Delta \xi \to 0}{\lim}G^{\prime }=G_{\infty }+\overset{M}{\underset{m=1}{\sum }}\frac{\eta_{m}}{\rho_{m}}=G_{0}$$

By replacing the stiffness tensor *C_ijkl_* with the time-dependent scalar *2G*, Equations (10) and (11) can be written as: (23)$$\Delta \sigma_{\alpha }^{R}=-\overset{M}{\underset{m=1}{\sum }}\left(1-e^{-\Delta \xi /\rho_{m}}\right)S_{\alpha_{m}}\left(\xi_{n}\right)$$
(24)$$S_{\alpha_{m}}\left(x_{k},\xi_{n}\right)=e^{-\Delta \xi /\rho_{m}}S_{m}\left(x_{k},\xi_{n}-\Delta \xi \right)+\eta_{m}2R_{\alpha }^{-1}R_{\varepsilon ,\alpha }\left(1-e^{-\Delta \xi /\rho_{m}}\right)$$

### 2.3. Adaptation to Homogenized Composite Behavior

A representative volume element (RVE) was set up, which consisted of two isotropic materials: a linear elastic fiber and a linear viscoelastic matrix. The fiber’s properties were chosen the same as for physical tests in Section 2, while the matrix was modeled by the method described in the previous section and the measured parameter values shown in Section 3. The compression modulus of the neat resin was chosen to be 2.78 GPa, derived from the elastic modulus and Poisson’s ratio in the elastic state.

The geometries of different textile structures and fiber volume fractions can be created efficiently by using the available material modelers of commercial FEM software. The incremental formulation of linear viscoelastic materials was implemented in a user material subroutine of Abaqus 6.13. It aimed to find a description for each element of the orthotropic relaxation tensor:(25)[σ1σ2σ3σ4σ5σ6]=[Ψ11Ψ21Ψ21Ψ12Ψ22Ψ23Ψ12Ψ23Ψ22 0Ψ44   Ψ44   Ψ66][ε1ε2ε3ε4ε5ε6]0

To do this, virtual load cases were set up, that created a unidirectional strain state. It consisted of 3024 elements of type C3D15, which is a 15-node quadratic triangular prism element. The fiber-matrix interface was modeled as perfectly bonded. In Figure 3, one can see the direction of the applied loads and the qualitative behavior of the set-up micromechanical models. To measure the response one could either evaluate the volume averaged stress or the reaction force. Here, as an indicator the reaction force was chosen (*R_f_*). The surface on which it was measured is also indicated in Figure 3. The boundary conditions are listed in Table A1 in Appendix B.

The resulting normalized values of the relaxation tensor at a temperature of 110 °C are shown in Figure 4. They can be compared with the normalized relaxation modulus of the neat resin, which is indicated by a dotted line. As shown by Meder et al. [35], some components appear to have an increasing value over time.

All these functions depend on one single time-dependent variable *G(t*). The decrease of each value has the same decay time, so all the curves can be expressed in terms of a scalar form factor and a master curve. Here, the form factor is defined as the ratio of the amount of normalized relaxation of the actual index and the neat resin reference at a reference time (Equation (26)). The respective elements of the relaxation tensor can then be described in terms of the elastic index, the form factor and the normalized relaxation modulus of the neat resin (Equation (27)):(26)$$f_{\alpha \beta }\left(t_{ref}\right)=\frac{1-{\bar{\Psi }}_{\alpha \beta }\left(t_{ref}\right)}{1-\bar{G}\left(t_{ref}\right)}$$
(27)$$\Psi_{\alpha \beta }\left(t\right)=C{\left(0\right)}_{\alpha \beta }{\bar{\Psi }}_{\alpha \beta }\left(t\right)=C_{\alpha \beta }\left(0\right)\left(\left(1-f_{\alpha \beta }\right)+f_{\alpha \beta }\bar{G}\left(t\right)\right)$$

Inserting this approach in Equation (6) (for the case of a relaxation test $C{\left(t\right)}_{\alpha \beta }=\Psi_{\alpha \beta }\left(t\right)$), the Prony series description of the stiffness tensor indices becomes: (28)$$C_{\alpha \beta }\left(\xi_{n+1}-\xi^{\prime }\right)=C_{\alpha \beta }\left(0\right)\left[\left(1-f_{\alpha \beta }\right)+f_{\alpha \beta }G_{\infty }+\overset{M}{\underset{m=1}{\sum }}f_{\alpha \beta }{\bar{G}}_{m}e^{-\left(\xi_{n+1}-\xi^{\prime }\right)/\rho_{m}}\right]$$

As in the isotropic case, this description of the stiffness tensor is inserted in the approach of Zocher [11] in Voigt notation. The respective form of Equation (A1) becomes:(29)$$\Delta \sigma_{\alpha }=C_{\alpha \beta }^{\prime }\Delta \varepsilon_{\beta }+\Delta \sigma_{\alpha }^{R}$$

Similar to the isotropic case, the first parameter on the right-hand side was approximated for small time steps without losing conservativity:(30)$$\underset{\Delta \xi \to 0}{\lim}C_{\alpha \beta }^{\prime }=C_{\alpha \beta }\left(0\right)$$

Since the decay time ρ does not depend on *β*, the sums in the adapted form of (1C) can be interchanged: (31)$$\Delta \sigma_{\alpha }^{R}=-\overset{6}{\underset{\beta =1}{\sum }}\overset{M}{\underset{m=1}{\sum }}\left(1-e^{-\frac{\Delta \xi }{\rho_{m}}}\right)S_{\alpha \beta_{m}}\left(\xi_{n}\right)\;=-\overset{M}{\underset{m=1}{\sum }}\left(1-e^{-\Delta \xi /\rho_{m}}\right)\overset{6}{\underset{\beta =1}{\sum }}S_{\alpha \beta_{m}}\left(\xi_{n}\right)$$

By comparing Equations (6) and (28), one can notice that only the part in the sum operator is time-dependent. It can be understood that Equation (11) changes to:(32)$$S_{\alpha \beta_{m}}\left(\xi_{n}\right)=e^{-\Delta \xi /\rho_{m}}S_{\alpha \beta_{m}}\left(\xi_{n}-\Delta \xi \right)+C_{\alpha \beta }\left(0\right)f_{\alpha \beta }{\bar{\eta }}_{m}R_{\varepsilon }\left(1-e^{-\Delta \xi /\rho_{m}}\right)$$

With the aim of reducing the number of state variables, an auxiliary state variable $S^{\prime }$ is defined. It consists of the sum over β of the initial state variables: (33)$$S_{\alpha_{m}}^{'}≔\overset{6}{\underset{\beta =1}{\sum }}S_{\alpha \beta_{m}}\left(\xi_{n}\right)$$

Equation (31) then takes the following form: (34)$$\Delta \sigma_{\alpha }^{R}=-\overset{M}{\underset{m=1}{\sum }}\left(1-e^{-\Delta \xi /\rho_{m}}\right)S_{\alpha_{m}}^{'}$$

By writing Equation (32) in terms of $S^{\prime }$, we obtain: (35)$$S_{\alpha_{m}}^{'}\left(\xi_{n}\right)=e^{-\Delta \xi /\rho_{m}}S_{\alpha_{m}}^{'}\left(\xi_{n}-\Delta \xi \right)+{\bar{\eta }}_{m}\left(1-e^{-\Delta \xi /\rho_{m}}\right)\overset{6}{\underset{\beta =1}{\sum }}C_{\alpha \beta }\left(0\right)f_{\alpha \beta }R_{\varepsilon ,\beta }$$

For small time steps, this can be approximated by: (36)$$S_{\alpha_{m}}^{'}\left(\xi_{n}\right)=e^{-\Delta \xi /\rho_{m}}S_{\alpha_{m}}^{'}\left(\xi_{n}-\Delta \xi \right)+\frac{{\bar{\eta }}_{m}}{{\bar{\rho }}_{m}}\overset{6}{\underset{\beta =1}{\sum }}C_{\alpha \beta }\left(0\right)f_{\alpha \beta }R_{\varepsilon ,\beta }\Delta \xi$$

We can then insert the relationship shown in Equation (12): (37)$$S_{\alpha_{m}}^{'}\left(\xi_{n}\right)=e^{-\Delta \xi /\rho_{m}}S_{\alpha_{m}}^{'}\left(\xi_{n}-\Delta \xi \right)+\frac{{\bar{\eta }}_{m}}{{\bar{\rho }}_{m}}\overset{6}{\underset{\beta =1}{\sum }}C_{\alpha \beta }\left(0\right)f_{\alpha \beta }\Delta \varepsilon_{\beta }$$

For the expression of the spring element of the normalized shear modulus, it is helpful to insert the parameter introduced in Equation (13). It represents the normalized amount of the decay described by one of the branches of the generalized Maxwell element, which was measured for the neat resin system. (38)$$S_{\alpha_{m}}^{'}\left(\xi_{n}\right)=e^{-\Delta \xi /\rho_{m}}S_{\alpha_{m}}^{'}\left(\xi_{n}-\Delta \xi \right)+\varphi_{m}\overset{6}{\underset{\beta =1}{\sum }}C_{\alpha \beta }\left(0\right)f_{\alpha \beta }\Delta \varepsilon_{\beta }$$

This method of describing the relaxation tensor indices with form factors allows reduction of the number of state variables of M × 6.

The concept presented here only works if there is no more than one individual material of the composite modeled to show the viscoelastic behavior. If there is more than one viscoelastic material included, the components of the relaxation tensor will have different decay times and cannot be transformed to a master curve with a form factor.

### 2.4. Generation of Experimental Data

The resin system investigated in this study was provided by Huntsman (Basel, Switzerland) and has the commercial name XB3585. It is a fast-curing epoxy resin for automotive applications that can achieve short curing-cycle times. The system has a final glass transition temperature of 120–130 °C and a flexural modulus of 2.8–3.1 GPa.

The glass fiber reinforcement textile was provided by Kümpers Composites (Salzbergen, Germany) as a unidirectional non-crimping fabric. The number of layers was chosen for a targeted fiber volume content of 60%. A summary of the number of layers, their orientation and the dimensions of the samples are presented in Table 1. Because of the limited maximum force of the test device (15 N), the specimens for the measurement in longitudinal dimensions were manufactured with a lower thickness. The textile consisted of S-glass fibers provided by 3B (Hoeilaart, Belgium). They are listed as SE 2020 2400 dTex and are expected to provide a tensile modulus of 81 GPa.

The relaxation of the specimens was measured with a 3-point bending test in a dynamic mechanical analyzer (DMA, Q800) from TA Instruments (New Castle, DE, USA) at different temperatures. The fully cured specimens were tested at isothermal temperatures ranging from 125 °C to 50 °C in steps of −5 °C. For each temperature step, an initial displacement of 0.1% strain was applied and the stress relaxation was recorded for 60 min. The relaxation modulus of the measured data according to the software’s internal manual was calculated as:(39)$$E\left(t\right)=K_{S}\left(t\right)\cdot \frac{L^{3}}{6I}\left[1+\frac{6}{10}*\left(1+v_{P}\right){\left(\frac{z}{L}\right)}^{2}\right]$$
where *K_s_* is the measured relationship between strain and force, *L* is the support width, *z* is the sample thickness, *I* is the moment of inertia and *v_P_* is Poisson’s ratio. The support width was chosen to be 50 mm.

### 2.5. Validation and Application on the Meso-Level/Ply Level

In order to validate the developed model, a virtual copy of the three-point bending test was built, whose simulation results were compared with the experimental data from the DMA. It consists of 896 elements of type C3D20R which is a 20-node quadratic brick. As for the micromechanical models, the incremental formulations of the transversely isotropic linear viscoelastic material behavior were implemented in a user subroutine. In Figure 5, the meshing and typical stress distribution during load application are shown. The derived relaxation stiffness of the virtual test was calculated as for the physical test using Equation (39) and the displacement and total reaction force. The supports were modeled by constraining the degree of freedom in a vertical direction. This is sufficient for the consideration here since the modeling of nonlinear effects of the supports is not intended.

## 3. Results

This section is structured to provide the best possible overview of the workflow carried out: First, the experimentally generated data of the three-point bending test are presented, divided into the measurement of the elastic modulus and the relaxation modulus. The data from the pure resin sample are used to calibrate the RVE model. Because this paper focuses on viscoelastic homogenization, the measured elastic modulus of the fiber-reinforced samples is used for the calibration of the validation model, while the measured relaxation modulus is used as the validation. In Section 3.3 the results of the RVE modeling are shown and the form factors are derived. This is followed by the validation in Section 3.4.

### 3.1. Experimentally-Derived Elastic Modulus

In Figure 6 the storage and loss modulus of the dynamic DMA measurements of the neat resin sample are presented. One can see that after an approximately linear decrease at lower temperatures, the storage modulus drops dramatically, starting at about 110 °C.

The corresponding result for the composite samples can be seen in Figure 7. While the storage modulus in the longitudinal direction remains stable until a temperature up to 115 °C, the modulus in transverse direction shows a similar behavior as the neat resin system. It is remarkable that the loss modulus of the material in the transverse direction shows an increase even at lower temperatures.

In the rheological model, which was used to describe the relaxation behavior of the neat resin, the elastic modulus is the sum of all springs of the generalized Maxwell model. In order to derive an analytical description, a customized expression was fitted to the measurement data. It consists of a linear part and a step function:(40)$$E^{\prime }=a+b\cdot x+\frac{c}{1+exp\left(\frac{T-d}{e}\right)}\;\left[MPa\right]$$

The parameters obtained after fitting this expression by linear regression to the measurement data are listed in Table 2.

The macroscopic model also requires a continuous description of the modulus of elasticity. For this purpose, the data of the composite samples was fitted in the same way. The resulting values are given in Table 2.

### 3.2. Experimentally-Derived Relaxation Modulus

The relaxation modulus of the neat resin system was measured at different constant temperatures from 50 to 125 °C. The isotropic viscoelastic description presented in Section 2.2 with two branches (M = 2) was fitted to the data using a linear regression algorithm. The measurement data and the result of the fitting are shown in Figure 8.

The fitting of the isotropic viscoelastic description resulted in a set of values for the parameters of the rheological model. They are shown in Figure 9. With increasing temperatures, the material approaches a rubbery and liquid state, and the decay time, indicated by the parameter $\rho_{i}$, decreases. Similarly, the amount of total relaxation increases, visible in the increase of ratio of spring elements connected to a branch with a damper element ($k_{1}/k_{tot}$ resp. $k_{2}/k_{tot}$). *k_tot_* denotes the sum of all springs in the rheological model, which was set to the same value as the modeled elastic modulus from the previous measurement.

To validate the macroscopic model later, the relaxation behavior of the composite material was measured. The result is shown in Figure 10. It can be seen that the relaxation modulus in the fiber direction shows hardly any relaxation up to about 100 °C. At higher temperatures, the relaxation modulus drops and reaches a level about 40% lower than the elastic value at room temperature. The relaxation modulus in the transverse direction shows a decrease at room temperature and continuously decreases at higher temperatures up to a reduction greater than 95% at 125 °C.

### 3.3. Results of RVE Homogenization

The normalized elements of the relaxation tensor ${\bar{\Psi }}_{\alpha \beta }$ obtained from the micromechanical model were compared with the behavior of the neat resin with Equation (26). The resulting form factors at a reference temperature of 110 °C are shown in Table 3.

In order to demonstrate the validity of the assumption that all elements ${\bar{\Psi }}_{\alpha \beta }$ can be expressed in terms of $\bar{G}$ and a scalar value $f_{\alpha \beta }$, the normalized relaxation tensor indices were transformed with Equation (41) and plotted together with the neat resin reference in Figure 11.
(41)$${\vec{\Psi }}_{\alpha \beta }\left(t\right):=\frac{{\bar{\Psi }}_{\alpha \beta }\left(t\right)-1}{f_{\alpha \beta }}+1$$

The largest relative deviation can be found for index ${\vec{\Psi }}_{44}$ with 20% followed by index ${\vec{\Psi }}_{66}$ with 10%. For all other indices, the maximum relative deviations are less than 2%.

### 3.4. Validation of Macroscopic Model

For an assessment of the quality of the relaxation model, the measured modulus and the modeled relaxation modulus derived from a three-point bending test were compared. The result is shown in Figure 12 and Figure 13 in normalized form. The comparison in normalized form was chosen in order to assess the quality of the relaxation model independent of the quality of the elastic modeling.

Data for temperatures above 115 °C were excluded, since the normalized measurement data do not show a trend towards increasing relaxation anymore. This can be explained by the measurement setup, for which the fixed sampling rate was not fast enough to capture the decay at the beginning of the measurement. This is indicated by the decrease in *k_2_/k_tot_* in Figure 9.

The modeling in the longitudinal direction is in a good agreement with the measurement data, while the modeling in the transverse direction can be seen to underestimate the amount of relaxation.

## 4. Discussion

Although the approach of describing the response of the individual elements of the stiffness tensor in terms of the behavior of the pure resin system and a scalar factor leads to an error of less than 2% for most elements, a larger deviation of up to 20% can be seen for the index ${\vec{\Psi }}_{44}$ and ${\vec{\Psi }}_{66}$. As a consequence, the model currently creates a maximum error for shear deformation of about the same order of magnitude. The deviation could be caused by the setup of the boundary conditions for this load case which is visible in the uneven surfaces of the 2–3 plane. This could be increased by the introduction of periodic boundary conditions as described by [25] in detail, where also a user-defined material subroutine in Abaqus was used.

The presented incremental method for the macroscopic time-dependent description of fiber composite materials represents an efficient alternative to established methods for FEM simulations. By introducing scalar form factors for the indices of the relaxation tensor, it allows the use of Mx6 state variables instead of M × 36 to describe an orthotropic material. This reduces the computational effort.

An increase of the accuracy of the model could be achieved by the introduction of temperature-dependent form factors. Furthermore, in order to derive a continuous description of the temperature dependency of the relaxation, the presented homogenization is able to introduce the time-temperature superposition principle with the use of the reduced time variable ξ. The time shift factors can be found by creating a master curve using the experimental data in Figure 8.

Linear viscoelastic material behavior is only an appropriate assumption for some polymers in the solid state at relatively low load levels. The higher the temperature, the more the state of the polymer changes from the glassy to the rubbery state. In measurements of the elastic modulus, an indicator of this change is the increase of the loss modulus. In the measurements taken here, this was the case for the neat resin at about 115 °C (Figure 6), for the composite in the longitudinal direction about the same (Figure 7), but in transverse direction it can be seen that there is an increase of the loss modulus already at lower temperatures which is an indicator for increased nonlinear behavior. This can be explained by the strain amplification effect on the microlevel, which reaches of the elastic limit of the material in some places at lower nominal load levels. One can assume that the strain amplification effect is more pronounced in the transverse direction, since there are stiffness jumps in the loading direction.

The strain amplification also changes the time-dependent behavior of the composite from linear viscoelastic to non-linear viscoplastic behavior. This was especially pronounced in the matrix-dominated directions (transverse), where the amount of relaxation in the measurement—especially at higher temperatures—exceeds the model’s prediction which assumes linear viscoelastic behavior. This means that the model provides a conservative assessment of the residual stresses, which is generally required for engineering purposes.

## 5. Conclusions

An incremental form to describe the linear viscoelastic response of isotropic and orthotropic materials was derived. Both forms were successfully implemented in an incremental FEM analysis.

With the use of a representative volume element it was shown that in the case of only one viscoelastic phase, the response of an element of the relaxation tensor can be expressed in terms of a scalar factor and the response of the viscoelastic phase. In this case, six form factors for a transversely isotropic material were derived. By inserting these factors into the incremental form for an orthotropic material, the macroscopic description of transversely isotropic material was converted to an extraordinarily easy form shown in Equations (29), (30), (34) and (38). Since the form factors describe the deviation of an index of the relaxation tensor of the composite in comparison with the neat resin, the impact of the virtual parameter derivation can be assessed easily, which is beneficial for engineering purposes. The result is a linear viscoelastic model with good handling and efficiency.

The model was validated through simulation of a three-point bending test and comparing the simulated results with the measurements in the longitudinal and transverse directions of a unidirectional fiber-reinforced material. A promising agreement was shown for up to 15 °C below the glass transition temperature. Since at these temperatures it was seen that most of the stress relaxes and the decay times are small, the present model was assessed to sufficiently describe the induced stresses of a typical manufacturing process of a thermoset material. Towards higher temperatures and, thus, increasing deviation of the model from the measured relaxation modulus, the model assessment still stays conservative due to the linear viscoelastic assumption, which is beneficial for component design.

## Figures and Tables

**Figure 1 materials-15-06724-f001:**
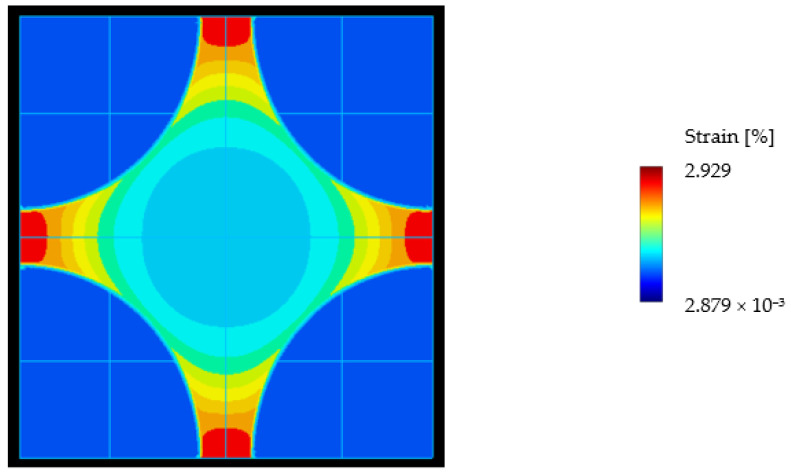
Von Mises strain distribution for the application of only thermal strain in the micromodel [37] (the blue part is the fiber).

**Figure 2 materials-15-06724-f002:**
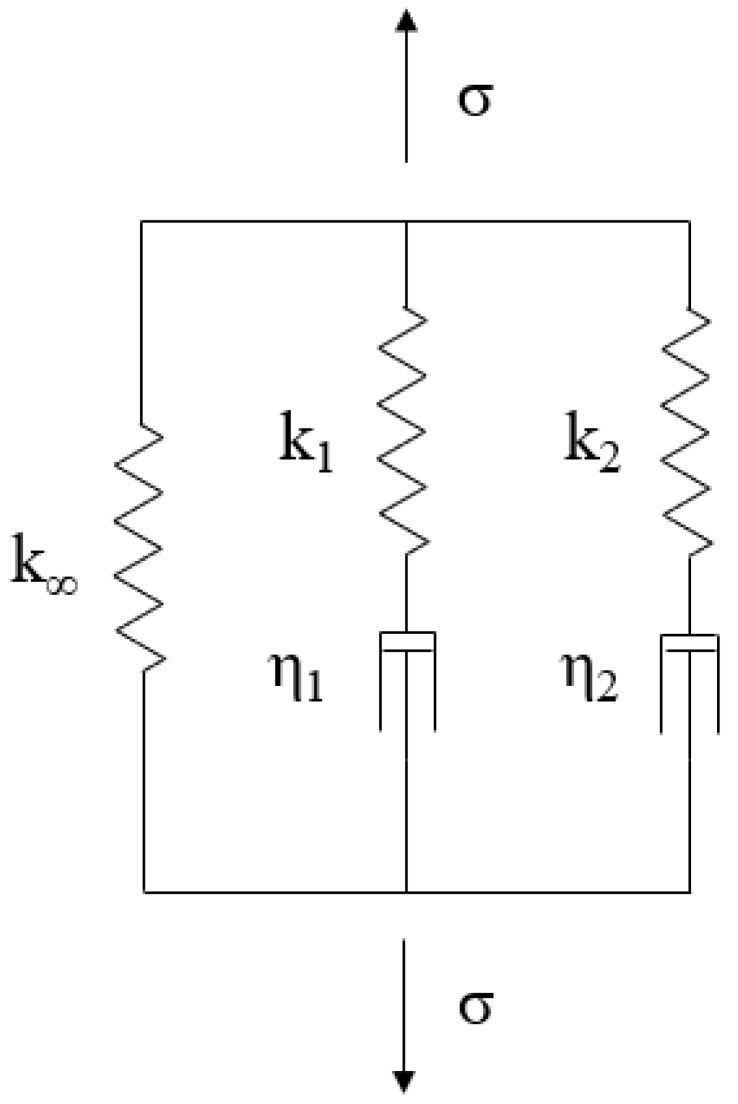
Generalized Maxwell model or Wiechert [38] model with M = 2 branches.

**Figure 3 materials-15-06724-f003:**
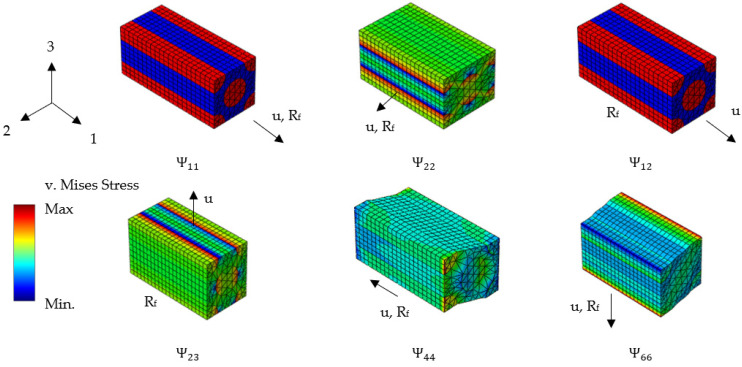
Definition of the boundary conditions for the derivation of the form factors defined in Equation (25).

**Figure 4 materials-15-06724-f004:**
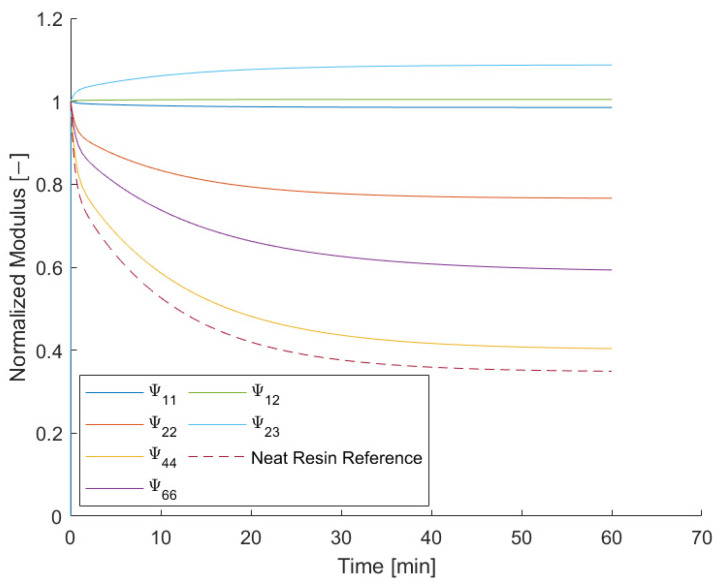
Normalized values of the entries of the relaxation tensor at 110 °C.

**Figure 5 materials-15-06724-f005:**
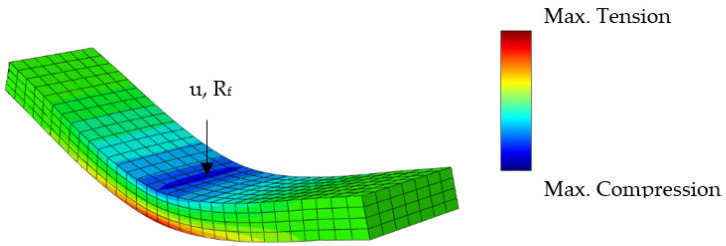
Geometry and qualitative stress distribution of the implemented validation model. Virtual copy of mechanical test setup. u denotes the applied deflection and *R_f_* the reaction force in the middle of the specimen.

**Figure 6 materials-15-06724-f006:**
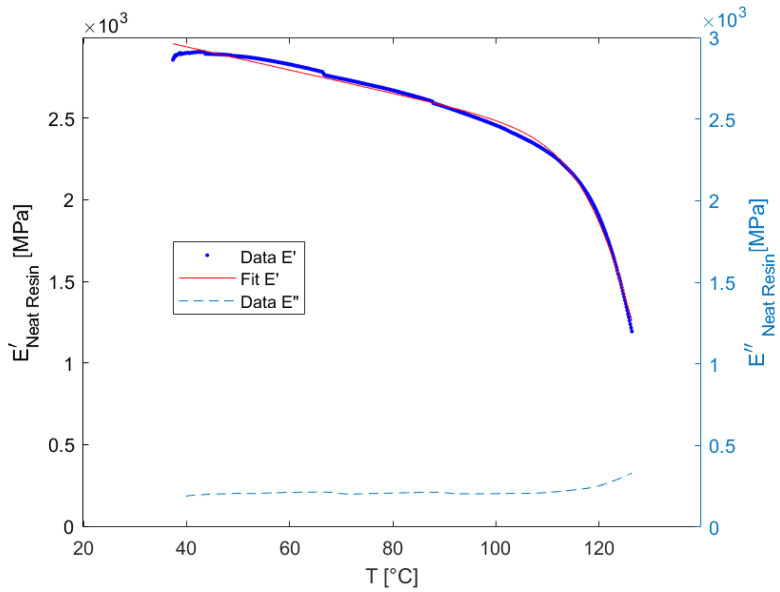
Measurement and fit of the storage and loss modulus of neat resin.

**Figure 7 materials-15-06724-f007:**
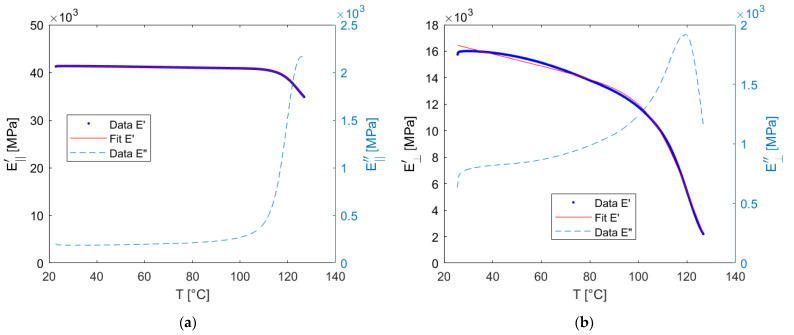
Measurement and fit of the storage and loss modulus of the UD-composite: (**a**) longitudinal; (**b**) transverse.

**Figure 8 materials-15-06724-f008:**
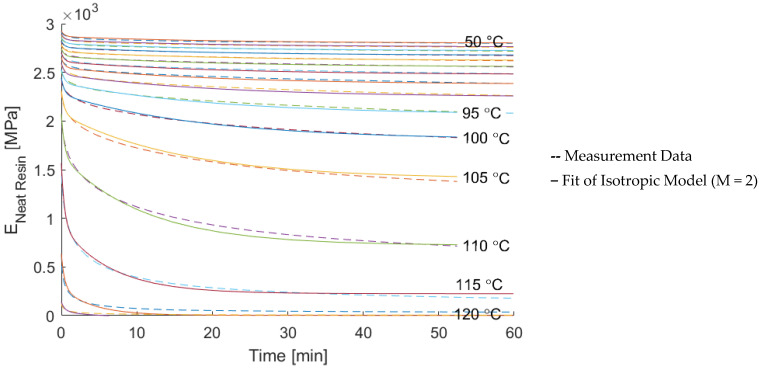
Comparison of the relaxation data and the relaxation model described by the continuous temperature-dependent parameters.

**Figure 9 materials-15-06724-f009:**
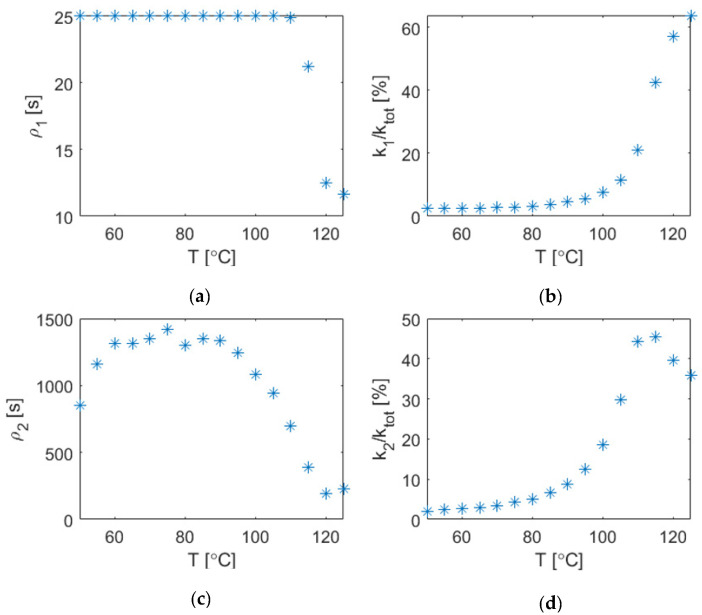
Resulting values of the element parameters of the rheological model after fitting to measurement data. It shows the decay time $\rho_{i}$ (**a**): branch 1, (**c**): branch 2 and the spring constant $k_{i}$ compared to the sum of all spring constants $k_{tot}$ (**b**): branch 1, (**d**): branch 2.

**Figure 10 materials-15-06724-f010:**
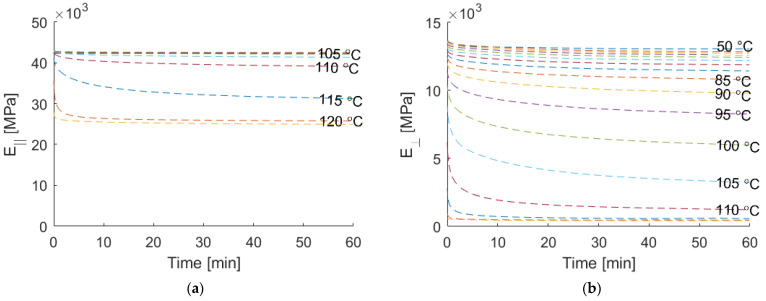
Measured relaxation modulus: (**a**) longitudinal direction; (**b**) transverse direction.

**Figure 11 materials-15-06724-f011:**
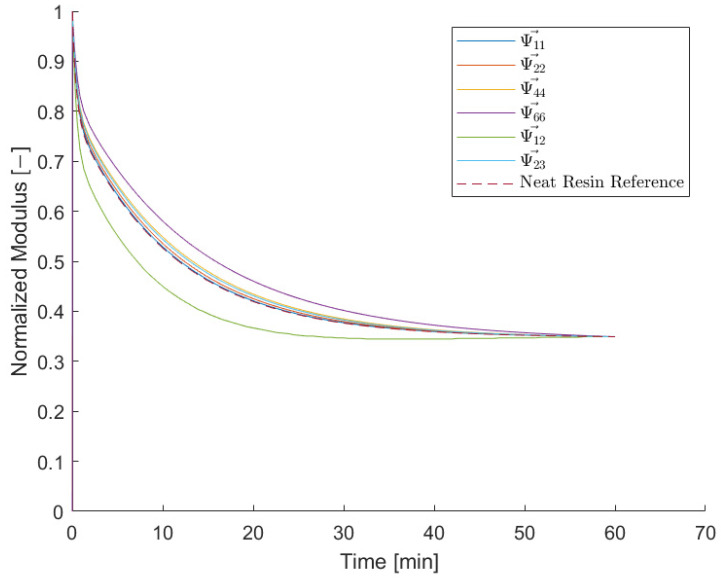
Relaxation tensor components transferred with the derived form factor. The transferred functions are in a good agreement with the reference function of the neat resin reference at 110 °C.

**Figure 12 materials-15-06724-f012:**
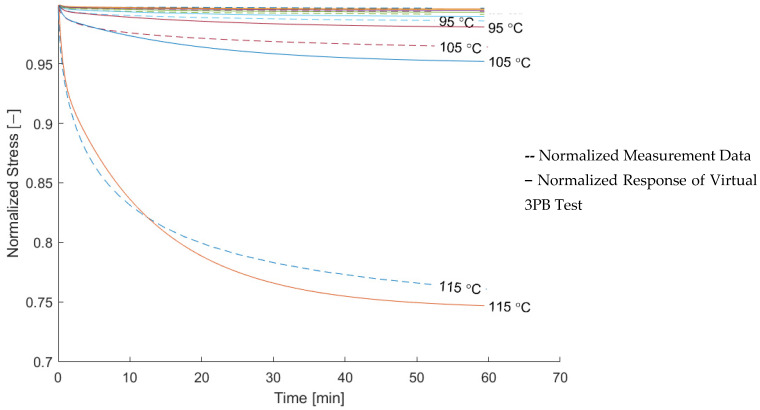
Comparison of the modeled and measured normalized relaxation modulus of the composite in the longitudinal direction.

**Figure 13 materials-15-06724-f013:**
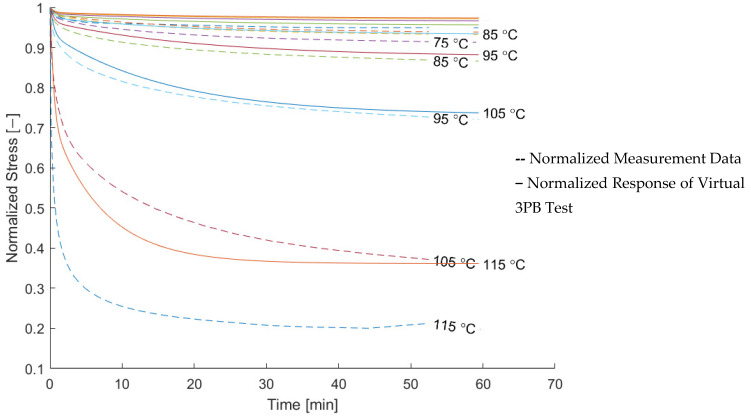
Comparison of the modeled and measured normalized relaxation modulus of the composite in the transverse direction.

**Table 1 materials-15-06724-t001:** Information of the samples.

Type	Thickness (mm)	Number of Layers	Width (mm)	Length (mm)
Neat resin	4 ± 0.06	-	15	65
Longitudinal	1 ± 0.04	2		
Transverse	4 ± 0.06	8		

**Table 2 materials-15-06724-t002:** Fitting parameters of the composite’s elastic modulus.

Direction	a	b	c	d	e
Neat	2.846 × 10^2^	−7.105	2.936 × 10^3^	1.301 × 10^2^	6.385
Longitudinal	3.176 × 10^4^	−6.417	9.770 × 10^3^	1.252 × 10^2^	4.027
Transverse	2.595 × 10^3^	−45.14	1.548 × 10^4^	1.217 × 10^2^	8.583

**Table 3 materials-15-06724-t003:** Results for the form factor f (110 °C).

Index	$f_{\alpha \beta }\left(t_{ref}\right)$
$\Psi_{11}$	2.307 × 10^−2^
$\Psi_{22}$	3.592 × 10^−1^
$\Psi_{44}$	9.161 × 10^−1^
$\Psi_{66}$	6.250 × 10^−1^
$\Psi_{12}$	6.453 × 10^−3^
$\Psi_{23}$	1.342 × 10^−1^

## Data Availability

The data presented in this study are available on request from the corresponding author.

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
