# Peer review of "Incremental Numerical Approach for Modeling the Macroscopic Viscoelastic Behavior of Fiber-Reinforced Composites Using a Representative Volume Element"

_materials, 2022, doi:10.3390/ma15196724_

Round 1

Reviewer 1 Report

The manuscript entitled “Incremental Numerical Approach for Modeling the Macroscopic Viscoelastic Behavior of Fiber-Reinforced Composites Using a Representative Volume Element” describes the stress-relaxation behavior of an epoxy-fiber composite. The given data and results are promising and match with the scope of Materials. I think that the manuscript can be accepted after addressing the following minor comments.

1- In Abstract, modify “The objective of this study was to describe” to “The objective of this study is ….”

2- In Introduction, the authors should add the full names before their abbreviations (e.g., RTM, C-RTM).

3- In Materials, identify the type of resin used, as vinylester, epoxy, polyester etc.

4- The Conclusion should be more concluded, providing the main findings.

Reviewer 2 Report

The authors describe a step-by-step method for modeling the macroscopic viscoelastic behavior of fiber-reinforced composites using a representative volume element. It is a fascinating study that deals with a current issue. The purpose of this study was to characterize the stress relaxation behavior of an epoxy-based fiber-reinforced material by modifying an existing incremental formulation of an orthotropic linear viscoelastic material behavior to Voigt notation and the specific case of an isotropic material. The clarity of purpose and methodology, as well as the soundness of the data, way of presentation, depth of discussion, and integrity of conclusions, distinguish this study. .However, the extent of application of the study's findings does not exist.

 As a result, I strongly propose that this manuscript be accepted for publication, but with the following drawbacks::-

1-      The application of this study must be included in the abstract section. Furthermore, the abstract should be brief and summarize the key findings.

2-      The study's originality should be addressed properly towards the end of the introduction section.

3-      Please mention whether the formulas used are from a reference, common equations such as Halpin-Tsai equations, or are developed by the authors.

4-      Please explain the virtual relaxation tests as well as the three bending tests (setup) used for model validation in the materials and methods section.

5-      Please include additional recommendations based on the results obtained in the conclusion section.

Reviewer 3 Report

This manuscript has been the focus on Incremental Numerical Approach for Modeling the Macroscopic Viscoelastic Behavior of Fiber-Reinforced Composites Using a Representative Volume Element. The study is original however I feel that the paper could be improved. Therefore, could you consider some points below for further improvement

1.           Methodology:

a)   The author should explain the flow of the research (Research Flow) in the context of the research methods (qualitative and quantitative) to help the reader understand the hypothesis for a specific process or phenomenon that has been observed during research.

2.           Results & discussion:

a)   Sections 3.1 and 3.2 - The authors should consider rearranging the tabulated data generated via the approach/method with more informatics arrangement (i.e., table of reduction factor/modulus strength).

b)  The paper is a train of tests, without explaining the outcome and the mechanistic reason behind observations. It is similar to a technical report frequently engineers use in production lines rather than a research article. What is the significant of the overall discussion? Please improve!!

3.           References:

a)     Please recheck the format references.

Reviewer 4 Report

REVIEW

This paper presents a computational approach for obtaining homogenized properties for a composite structure modeled as an orthotropic linear viscoelastic material. The method employs the finite element method to evaluate the response of a unidirectional continuous fiber composite under various loadings where the compute stress-strain relationship is used to compute constants for the homogenized viscoelastic model. The fiber is assumed to be elastic while the matrix is modeled as an isotropic linear viscoelastic solid where properties are obtained via creep testing. Stresses computed in the fiber direction with the proposed homogenized model match well with measured data, however, the prediction of transverse stresses are not as good.

The paper offers an interesting approach to computing equivalent homogenized properties for viscoelastic solids, however, there are several concerns that should be addressed through a resubmission before the work is published. These are included in the following:

1) As mentioned above, the transverse results are quite poor in comparison to the longitudinal predictions (see Figures 12 and 13). From these curves, it appears that the method may be limited to longitudinal modulus and stress predictions. If this is the case, then the value of the approach should be well justified given that the composite considered is a continuous fiber composite with an elastic fiber.

2) Wang and Smith (Composite Structures, 229, 2019) proposed a similar approach but is not referenced here. Their approach appears to be able to predict transverse moduli and is shown to work on the more difficult problem of aligned short fiber composite.

3) It has been documented that periodic boundary conditions provide more accurate RVE results than prescribed boundary conditions as shown in this paper. A strong justification is needed here if the work is to be published without using periodic boundary conditions.

4) There needs to be more attention given to the literature on using finite elements and RVEs to compute effective properties. This paper has very little background on FEA methods for doing these computations.

5) The plots in Figure 11 lack specificity in assessing the error in the proposed approach. A quantitative error measure would be helpful. As shown, it is difficult to draw the conclusion in the statement immediately following Figure 11.

6) Curves are labeled as ‘Data’ in figures 12 and 13, but it is not clear where this information is obtained.

7) Curves in Figures 7 and 10 should be individually labeled or a legend should be provided.

8) There are curves in Figures 6 and 9 that do not appear in the legend.

9) The paper uses the words ‘This’ and ‘These’ extensively throughout and it is mostly not clear what it is referring to.

Round 2

Reviewer 2 Report

It is suggested that the work be accepted in its present form because the authors have successfully taken into consideration the reviewers' comments and observations.